# The Effect of Knowledge, Attitudes, and Practices of Korean Correctional Officers about COVID-19 on Job Stress

**DOI:** 10.3390/ijerph18157779

**Published:** 2021-07-22

**Authors:** Hyun-Ok Jung, Seung-Woo Han

**Affiliations:** 1Department of Nursing, College of Nursing, Kyungpook National University, Daegu 41944, Korea; juiris@korea.kr; 2Department of Emergency Medical Technology, Kyungil University, Gyeongsan 38428, Korea

**Keywords:** correctional institution, government official, job stress, knowledge, attitudes, practices, COVID-19

## Abstract

The purpose of this study was conducted to investigate the effects of corrective officers’ knowledge, attitudes, and practices on job stress. The subjects of this study were 375 randomly selected male correctional officials working at five South Korean correctional facilities that had been affected by COVID-19. This study considered data collected with approval from 17 May 2021 to 14 June 2021. Knowledge, attitudes, practices, and job stress in relation to COVID-19 were assessed using a personal questionnaire. The data were analyzed using mean, standard deviation, t-test, one-way ANOVA, and post-test using Pearson’s correlation coefficient. The job stress of participants was negatively correlated with knowledge, attitudes, and practices. Significant factors influencing job stress included knowledge and practices. These factors explained 38% of the variance. In this study, knowledge and practices were identified as influencing the job stress of correctional officers. These results are intended to contribute to the development of programs that can enhance the COVID-19-related knowledge and practices of correctional officers and reduce job stress.

## 1. Introduction

Coronavirus disease (COVID-19) is a contagious and potentially lethal viral disease [1]. It is transmitted very strongly by respiratory aerosols or direct contact [2], so the main environmental factors of infection are considered to be density, confinement, and closeness, which are also characteristics of correctional facilities such as prisons. Over the past year, at least one case of COVID-19 has occurred in 19 of the 52 correctional facilities in Korea, and an epidemiological investigation and intensive quarantine were conducted. In particular, 1068 COVID-19 patients were confirmed at one detention center that accommodated 2549 people, and the highest level of social distancing measures was implemented in the national correctional facilities [3].

Korean correctional officials experience various job stresses as a result of the closed correctional environments and their interactions with prisoners physically, mentally, administratively, and legally [4]. During the period when the highest level of social distancing was mandated in national correctional facilities, all employees were prohibited from engaging in external activities that were not related to the emergency work system, prisoners were not able to meet face-to-face, and prisoners’ activities, such as work and education, in the correctional facilities were completely suspended [3].

People who directly respond to infectious diseases can develop mental health conditions such as post-traumatic stress disorder, depression, insomnia, and severe anxiety due to high levels of job stress. These potential psychological negative consequences reduce an individual’s ability to effectively handle health emergencies, as well as impacting their well-being [5].

COVID-19 has no specific treatment, only a preventive vaccine [6]. Control of the spread of COVID-19 requires preventive measures that people can implement easily in their daily lives. Thus far, the health authorities in Korea have prepared preventive action guidelines to prevent the spread of COVID-19, selected the priority group to which the guidelines should be applied, and identified the importance of the preventive action of the group [7]. One such group is composed of correctional facilities such as prisons and detention centers, where correctional officers and prisoners live together. Risk management of non-treatable infectious diseases, such as COVID-19, is important to the extent to which people comply with the health authority’s recommendations. This crisis management is heavily influenced by people’s knowledge, attitudes, and practice [6].

The knowledge, attitudes, and practices (KAP) in relation to COVID-19 have been found to affect a society’s readiness to accept behavioral change measures from health authorities [8]. A high level of knowledge and attitudes related to COVID-19 is positively associated with preventive actions [9,10,11,12]. However, knowledge and attitudes may further complicate measures to curb panic, emotional levels, and disease spread [13].

Correctional officers have not been targeted in previous studies of knowledge, attitudes, and behaviors regarding COVID-19; looking at prior studies, we investigated the knowledge, attitudes, and practices regarding COVID-19 in general adults [14,15,16,17], university students [7,8,12], and medical workers [9,10,11,18] based on the guidelines presented by the World Health Organization (WHO). Therefore, this study will investigate the effects of the knowledge, attitude, and behavior in relation to job stress among high-risk correctional officers in an environment where COVID-19 is prevalent, and then seek ways to reduce job stress among correctional officers. Furthermore, we seek to promote the efficiency of the main tasks of correctional officials (i.e., management of the detention and correction of prisoners).

## 2. Materials and Methods

### 2.1. Research Design

This study involved descriptive research using a self-reported survey method to understand how correctional officers’ knowledge, attitudes, and behaviors related to COVID-19 are associated with their job stress.

### 2.2. Study Subjects

The subjects of this study were randomly selected among male correctional officers working at five South Korean correctional facilities that have experienced outbreaks of COVID-19. The number of subjects was calculated by including 14 independent variables based on α = 0.05, effect size d = 0.10, power value = 0.95 according to the G*Power 3.1 program, and the minimum number of samples required for multiple linear regression was 236. Considering the potential dropout rate, 400 copies of the questionnaire were distributed in this study. Among them, a total of 375 subjects were selected; 25 were rejected because they left many questions unanswered or gave unfaithful responses.

Specific criteria for selecting eligible subjects were as follows: (1) a correctional officer or prisoner who was diagnosed as positive for COVID-19 within the last 6 months and is working at an institution that conducted an epidemiological investigation through a public health center, (2) those who are currently working at an institution that has conducted or is currently conducting an intensive quarantine for 4 weeks or more, (3) those who voluntarily agreed to participate in the study after hearing the explanation of this study, (4) those who can read and communicate in Korean.

### 2.3. Research Tools

#### 2.3.1. Knowledge, Attitudes, and Practices Regarding COVID-19

COVID-19-related KAP measures developed elsewhere [15] were adapted and modified for this study. The measure consisted of a total of 40 questions: 17 questions about knowledge, 10 questions about attitudes, 13 questions about behavior. All assessments used Likert scales. For the knowledge domain (e.g., ‘What are the transmission routes of COVID-19?’), the scale had four points (4 = ‘strongly agree’ to 1 = ‘strongly disagree’); for the attitude domain (e.g., ‘To what extent do you observe the preventive measures against COVID-19?’), the scale had five points (5 = ‘very high’ to 1 = ‘very low’); and for the practice domain (e.g., ‘How often do you use a face mask?’), the scale had five points (5 = ‘always’ to 1 = ‘never’). In the previous study [15], the Cronbach’s α scores for the domains were 0.64, 0.78, and 0.82, respectively, and in this study, they were 0.88, 0.84, and 0.71, respectively.

#### 2.3.2. Job Stress

The basic type of Korean job stress measurement tool [19] was used. The measure had a total of 43 questions, consisting of eight domains: physical environment, job demand, efficient job control, interpersonal confidence, job security, occupational system, lack of reward, and occupational climate. This tool used a four-point Likert scale (from 1 = ‘very much’ to 4 = ‘not at all’), and 18 questions were inverse questions. In this tool, a high score indicated high job stress. In the previous study [19], the Cronbach’s α score for the subdomains ranged from 0.51 to 0.82, and in the present study, they ranged from 0.50 to 0.86, with an overall α = 0.79.

### 2.4. Data Collection

This study was approved by the Research Ethics Committee (IRB) of K University in Korea (IRB NO. 1041459-202105-HR-008-01), to protect the subjects and obtain justification for the research progress. Data were collected from 17 May 2021 to 14 June 2021, according to the approved contents. To generalize the study, we visited five correctional facilities where COVID-19 outbreaks had occurred, explained the purpose and method of the study to the person in charge of the psychotherapy team, and received permission and cooperation to conduct the study. We explained to them that the survey results would be processed numerically and would not be used for any purpose other than the research. In addition, we explained that the subjects’ anonymity would be maintained and that they could stop participating in the survey at any time while completing it. We obtained the written consent of the subjects to participate in the study, and then asked them to fill out the questionnaire individually. The researchers collected the questionnaires directly.

### 2.5. Data Analysis

The collected data were analyzed using the IBM SPSS statistics 23.0 software (IBM Corp, Armonk, NY, USA). The distribution of subjects’ general characteristics was analyzed as numbers and percentages, and the difference test of job stress according to subjects’ general characteristics was analyzed using a t-test and one-way ANOVA. The degree of job stress and knowledge, attitudes, and practices related to COVID-19 were expressed as average and standard deviation. Correlations between job stress and related variables were determined using Pearson’s correlation coefficients. The assumptions of regression analysis were verified by testing the autocorrelation and multicollinearity among the effects of independent variables, and by inspecting the residuals, and then multiple linear regression was used to determine the effect factors on job stress.

## 3. Results

### 3.1. Distribution by General Characteristics of Subjects

The most common answers to each question were as follows: 60.8% (228 people) were 40 years or older; 73.3% (275) were college graduates; 54.7% (205) had no religion; 61.3% (230) were married; 46.4% (174) had <10 years of tenure; 33.9% (127) were seventh-grade correctional officers; 86.9% (326) had no chronic diseases; 53.9% (202) were generally healthy; 92.5% (347) had no respiratory diseases; 72.5% (272) had not been vaccinated against COVID-19, and 84.5% (317) had had COVID-19-related education.

Subsequent analyses confirmed differences in job stress according to the general characteristics of subjects’ marriage status and tenure (Table 1). Job stress was significantly higher among those over 40 (106.43 ± 9.24) compared to those under 40 (104.05 ± 8.35) (t = −2.53, *p* = 0.012), among married (106.23 ± 9.18) compared to unmarried (104.24 ± 8.46) employees (t = −2.09, *p* = 0.037), and in those who had 20–30 years of tenure (108.62 ± 8.44) compared to those who had < 10 years (104.23 ± 8.14) (F = 5.26, *p* < 0.001) (Table 1).

### 3.2. Degree of Job Stress, Knowledge, Attitudes, and Practices

Regarding COVID-19, the subjects’ degree of knowledge averaged 50.80 ± 6.54 points (range 33.0 to 68.0), the degree of attitude averaged 38.04 ± 5.46 points (range 16.0 to 50.0), the degree of practice averaged 42.29 ± 5.68 points (range 26.0 to 65.0), and the degree of job stress averaged 105.50 ± 8.97 points (range 55.0 to 132.0) (Table 2).

### 3.3. Correlation between Study Variables

Knowledge, attitudes, and practices were negatively correlated with each other. Job stress had a statistically significant negative correlation with knowledge (r = −0.319, *p* < 0.01), attitudes (r = −0.127, *p* < 0.05), and practices (r = −0.237, *p* < 0.01) (Table 3).

### 3.4. Factors That Affect Job Stress

To identify the factors that affected the subjects’ job stress, a multiple regression analysis was performed using as explanatory variables age, marriage, and tenure, which all showed significant differences in terms of the demographic characteristics. The Durbin–Watson test statistic was 1.691, which is close to 2, which indicates no significant autocorrelation, and the tolerance was 0.343 to 0.765, which exceeds the value of 0.1, below which multicollinearity can be declared. The regression model of this study was significant (*p* < 0.001), and had a coefficient of determination R^2^ = 0.38, which means that it explains 38% of the variation in the data. Knowledge (β = −0.268, *p* < 0.001) and practices (β = −0.156, *p* = 0.005) had statistically significant associations with the subjects’ job stress, i.e., job stress decreased as the level of knowledge or practices increased (Table 4).

## 4. Discussion

Recently, Korea has been experiencing increased public malaise due to the prolonged COVID-19 outbreak and is experiencing unpredictable long-term crises and difficulties. The government of Korea is controlling the epidemic by strengthening the containment policy to prevent the spread of COVID-19, but it is still struggling with the high contagiousness of the virus and its rapid spread. In particular, the outbreak of COVID-19 in densely populated areas such as prisons, the military, and schools can spread the disease, and high numbers of cases could cause the collapse of the medical system. The rapid spread of COVID-19 has been halted, but the number of confirmed cases continues to rise, so concerns remain. Therefore, this study intends to prepare basic data to find ways to change the job stress of correctional officials in a positive direction by identifying the effects of knowledge, attitudes, and behavior related to COVID-19 on job stress.

The analysis identified that an increase in knowledge about COVID-19 was correlated with a decrease in self-reported job stress. High levels of job stress can degrade an employee’s job performance [20]. The obvious conclusion is that efforts to increase the level of knowledge about COVID-19 will help the employees to manage their job stress, and this will improve job quality. This finding is consistent with an earlier research result [21] that an increase in the level of job knowledge decreases job stress. Correctional officers experience various job stresses due to their interactions with prisoners, physically, mentally, administratively, and legally [4], and the increase in prisoner management tasks due to COVID-19 will cause an increase in job stress. However, if the level of job-related COVID-19 knowledge increases, the employee’s ability to control his work environment will improve, so his ability to manage his job stress and reduce tension will improve, as will his job performance [22]. Therefore, to improve job performance, ways to improve employees’ knowledge about COVID-19 at the national level must be found by creating a work environment that can improve work-related knowledge about COVID-19 in special circumstances such as the outbreak of infectious diseases.

Practices also had a significant correlation with the job stress of the correctional officers. This result is consistent with the results of a previous study [23], which showed that job stress had a significant influence on nurses’ patient safety activities. In a situation in which COVID-19 is spreading nationwide, the correctional officers’ degree of knowledge of the COVID-19-related rules of conduct, and of ways to implement them, will help to reduce the spread of infection among inmates and achieve safe management of the prison environment. Correctional officers faced situations where they had to prevent the spread of infections and maintain their job performance as an extension of their busy daily duties, such as by monitoring how often they washed their hands and maintaining an acceptable social distance. This observation shows that excessive job stress also affects other aspects of the job environment, such as by degrading the physical working conditions and causing an excessive workload or time pressure [24]. However, another previous study [1] reported that COVID-19 prevention rules through appropriate training had an effect of reducing the spread of COVID-19 infection. These findings suggest that the practice of preventive actions may be initially considered a burden, as an extension of an already heavy workload, but will improve the work environment and reduce job stress by preventing infection in the organization and providing an environment that is safe from COVID-19. Therefore, future research should identify the effects of various behavioral rules on workload and burnout in the field, and identify behavioral factors that can actually affect job stress.

Lastly, the employees’ attitudes did not display a significant correlation with job stress. A previous study [9] found that an increase in negative attitudes had a negative impact on the organization, by increasing job stress and damaging interpersonal relationships [25]. However, in the present study, social awareness may have been insufficient for subjects to have developed a complete attitude toward COVID-19 in the field. This is the first experience of COVID-19 in Korean society, so it may take time for the new circumstances to change the attitudes of the correctional officers. Accordingly, a future study should test a previous conclusion [26] that the increase in inappropriate service attitude has a strong effect on job stress, and that various program interventions and counseling techniques should be developed that can improve job attitudes and reduce job stress.

Considering the results of the discussion so far, the lower the knowledge and practices of correctional officers related to COVID-19, the higher the level of job stress. Therefore, within the crisis management of COVID-19, it is necessary to improve the COVID-19-related work environment for correctional officers and to increase their knowledge and practices related to COVID-19.

A limitation of this study is that the participants were arbitrarily sampled from male correctional officers working in five correctional facilities in Korea, so its conclusions may not be generalizable to all correctional facilities in Korea. Therefore, future research should verify the knowledge, attitudes, and practices related to COVID-19 among all correctional facilities in Korea, and to seek ways to reduce job stress and systematically improve the work environment. Moreover, this study found that demographic characteristics did not affect job stress. However, a previous study [27] of nurses exposed to COVID-19 found that age and working hours per week were demographic variables that affected job stress, and another study [28] of teachers exposed to COVID-19 showed that gender, age, and income level affected job stress. Future research should explore and test practical demographic variables that are suitable for the characteristics of correctional officers. Lastly, as interest in COVID-19 has risen, many quantitative studies have been conducted based on various subjects, but the ability to interpret the real job stress experienced as a result of COVID-19 may be limited. Future research should use a multifaceted, qualitative approach that uses interviews to identify job stress-related experiences.

## 5. Conclusions

This study aimed to understand the effects of correctional officers’ knowledge, attitudes, and practices related to COVID-19 on job stress. This study identified knowledge and practices as factors that affected the job stress of correctional officers. Therefore, the purpose of this study was to provide basic data on measures that can positively overcome the COVID-19 crisis by improving the COVID-19-related knowledge and behavior of correctional officers to help reduce their job stress and improve their work performance.

## Figures and Tables

**Table 1 ijerph-18-07779-t001:** Distribution by general characteristics of subjects.

Variable	Category	Frequency	M(SD)	t/F	*p*
Age	20–39 years old	147(39.2)	104.05(8.35)	−2.53	0.012
	≥40	228(60.8)	106.43(9.24)		
Educational background	High school graduate	78(20.8)	105.56(9.08)	0.13	0.877
	College graduate	275(73.3)	105.55(8.88)		
	Graduate school or higher	22(5.9)	104.55(9.96)		
Religion	Have	170(45.3)	105.49(8.57)	−0.15	0.988
	Not have	205(54.7)	105.50(9.30)		
Marriage	Unmarried	145(38.7)	104.24(8.46)	−2.09	0.037
	Married	230(61.3)	106.23(9.18)		
Working Period	Less than 10 years	174(46.4)	104.23(8.14)	5.26	0.001
	10–20 years	93(24.8)	104.61(9.83)		
	20–30 years	77(20.5)	108.62(8.44)		
	More than 30 years	31(8.3)	107.48(10.15)		
Position	Prison officer	91(24.3)	104.22(9.01)	2.14	0.094
	9th grade correctional officer	84(22.4)	104.27(8.38)		
	7th grade correctional officer	127(33.9)	106.60(9.65)		
	Governor of a prison	73(19.5)	106.58(8.07)		
Chronic disease	Have	49(13.1)	105.49(8.83)	−0.03	0.977
	Not have	326(86.9)	105.53(9.90)		
Health status	Healthy	202(53.9)	104.70(9.40)	1.85	0.159
	Normal	164(43.7)	106.34(8.26)		
	Not healthy	9(2.4)	107.89(10.42)		
Respiratory diseases	Have	28(7.5)	103.32(10.46)	1.34	0.183
	Not have	347(92.5)	105.67(8.83)		
Vaccination	Vaccinated	103(27.5)	105.27(9.60)	0.30	0.766
	Not vaccinated	272(72.5)	105.58(8.73)		
COVID-19 Education status	Yes	317(84.5)	105.35(8.97)	0.74	0.462
	No	58(15.5)	106.29(8.96)		

**Table 2 ijerph-18-07779-t002:** Degree of job stress, knowledge, attitudes, and practices of the subjects.

	M(SD)	Min	Max
Knowledge	50.80(6.54)	33.0	68.0
Attitude	38.04(5.46)	16.0	50.0
Practice	42.29(5.68)	26.0	65.0
Job stress	105.50(8.97)	55.0	132.0

**Table 3 ijerph-18-07779-t003:** Correlation between study variables.

	Job Stress	Knowledge	Attitude	Practice
Job stress	1			
Knowledge	−0.319(<0.01)	1		
Attitude	−0.127(<0.05)	0.411(<0.01)	1	
Practice	−0.237(<0.01)	0.386(<0.01)	0.408(<0.01)	1

**Table 4 ijerph-18-07779-t004:** Factors affecting job stress.

	β	SE	t	*p*
Age	−0.015	0.803	−0.178	0.859
Marriage	0.035	1.095	0.547	0.584
Tenure	0.152	0.723	1.888	0.060
Knowledge	−0.268	0.075	−4.871	0.001
Attitude	0.045	0.093	0.791	0.430
Practice	−0.156	0.087	−2.828	0.005

Durbin–Watson = 1.691, Tolerance Limit = 0.343∼0.765, VIF = 1.307∼2.917 R^2^ = 0.38, Adjusted R^2^ = 0.38, F = 10.42, *p* < 0.001.

## Data Availability

Data is contained within the article.

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
