# Peer review of "The Effect of Knowledge, Attitudes, and Practices of Korean Correctional Officers about COVID-19 on Job Stress"

_ijerph, 2021, doi:10.3390/ijerph18157779_

Round 1
Reviewer 1 Report
The work investigates a specific and interesting issue using, for the first time, the KAP model. I report below some suggestions.
ABSTRACT
“considered data collected data”: probably to be fixed
“conclusion: In this study,” : I think the abstract should be unstructured
INTRODUCTION
I suggest to provide a brief explanation of the KAP model
“COVID-19 has no specific treatment, only a preventive vaccine.” Please, provide a reference
“Furthermore, it is intended to promote efficiency of the main tasks of correctional officials; i.e., management of the detention and correction of prisoners.” Add parenthesis before “i.e.” and after “prisoners”
MATERIALS AND METHOD
“This study was descriptive research using a self-reported survey method to under-stand how correctional officials' knowledge, attitudes, and behaviors related to COVID-19 affect their job stress.” As the study cannot demonstrate causal relationships, I suggest to change “affect” with something like “is associated with”.
“15 were rejected because the left” Change “the” with “they”
“Specific criteria for selecting eligible subjects are as follows” I suggest to be consistent with verbal tenses throughout the manuscript. For instance, I would use “were” instead of “are”.
“All assessments used Likert scales; in each, a high level” What do they mean with “high level”?
RESULTS
“Knowledge (β = -0.268, p < 0.001) and practice (β = -0.156, p = 0.005) had statis-tically significant effects on the subject's job stress; i.e., job stress decreased as the level of knowledge or practice increased” I would talk about “associations” rather than “effects”
Author Response
Response to Reviewer 1 Comments
ABSTRACT
Point 1: “considered data collected data”: probably to be fixed
Response 1: This study considered data collected with approval from 17 May 2021 to 14 June 2021.
Point 2: “conclusion: In this study,” : I think the abstract should be unstructured
Response 2: Conclusion: In this study~
INTRODUCTION
Point 3: I suggest to provide a brief explanation of the KAP model
Response 3: The text is described as a model developed by the World Health Organization (WHO) as a measure of knowledge, attitude, and practice against COVID-19. I have added something related to this.
“Risk management of non-treatable infectious diseases, such as COVID-19, is important to the extent to which people comply with the health authority's recommendations. This crisis management is heavily influenced by people's knowledge, attitudes and practice [6].”
“Looking at prior studies, we investigated knowledge, attitudes, and practice of COVID-19 in general adults [14-17], university students [7,8,12] and medical workers [9-11,18] based on the guidelines presented by the World Health Organization (WHO).”
Point 4: “COVID-19 has no specific treatment, only a preventive vaccine.” Please, provide a reference
Response 4: COVID-19 has no specific treatment, only a preventive vaccine [6].
Point 5:“Furthermore, it is intended to promote efficiency of the main tasks of correctional officials; i.e., management of the detention and correction of prisoners.” Add parenthesis before “i.e.” and after “prisoners
Response 5: Furthermore, it is intended to promote efficiency of the main tasks of correctional officials; (i.e., management of the detention and correction of prisoners).
MATERIALS AND METHOD
Point 6: “This study was descriptive research using a self-reported survey method to under-stand how correctional officials' knowledge, attitudes, and behaviors related to COVID-19 affect their job stress.” As the study cannot demonstrate causal relationships, I suggest to change “affect” with something like “is associated with”.
Response 6: This study was descriptive research using a self-reported survey method to understand how correctional officials' knowledge, attitudes, and behaviors related to COVID-19 are associated with their job stress.
Point 7: “15 were rejected because the left” Change “the” with “they”
Response 7: because they left many questions unanswered, or gave unfaithful responses.
Point 8: “Specific criteria for selecting eligible subjects are as follows” I suggest to be consistent with verbal tenses throughout the manuscript. For instance, I would use “were” instead of “are”.
Response 8: Specific criteria for selecting eligible subjects were as follows:
Point 9: “All assessments used Likert scales; in each, a high level” What do they mean with “high level”?
Response 9: This is my mistake. I modified the sentence. “All assessments used Likert scales; in each, a high level.”
RESULTS
Point 10: “Knowledge (β = -0.268, p < 0.001) and practice (β = -0.156, p = 0.005) had statis-tically significant effects on the subject's job stress; i.e., job stress decreased as the level of knowledge or practice increased” I would talk about “associations” rather than “effects”
Response 10: Knowledge (β = -0.268, p < 0.001) and practice (β = -0.156, p = 0.005) had statistically significant associations on the subject's job stress; i.e.,

Reviewer 2 Report
The study investigated the link between Knowledge, Attitudes and Practices (KAP) of male correctional officers with their level of job stress during COVID-19 outbreak in Korea. Results indicate that job stress is negatively correlated with the three KAP constructs, whereas – in a multiple regression model – unique negative associations emerge with Knowledge and Practices.
Overall, the study is easy to follow and provides some exploratory insights into the determinants of job stress for a potentially neglected population. I have some (minor and major) suggestions and comments.
Abstract:
- I advise to include the sample size in the abstract.
- Please correct the expression “data collected data”.
Introduction:
It is globally a good and concise introduction to the methods section. The authors may want to consider additional literature to cite, as there are several studies conducted in Western countries too.
- For instance, when mentioning preventive behavioural measures for COVID-19 mitigation (end of p. 1), you may consider citing Costantini et al. (2021; doi: 10.3389/fpsyg.2021.635406): the authors also provide some information regarding the impact of knowledge on mitigation behaviours, which may be of interest for your discussion. There are also studies investigating the impact of COVID-19 on psychological distress. I will mention some, including Viviani et al. (2021; https://doi.org/10.1016/j.future.2021.06.044), Preti et al. (2020a; https://doi.org/10.1007/s11920-020-01166-z) and Preti et al. (2020b; https://doi.org/10.1007/s11920-020-01204-w): these studies/reviews demonstrate that COVID-19 has an impact on the mental health of laypersons and specific populations, therefore they may serve as an introduction to the rationale of your own investigation.
Methods:
There are a few improvements I think should be made to the methods section.
- The authors mention G-power, but no description of their power analysis is provided. For instance: A) Was it a priori? B) What kind of model did you perform the power on? A regression model? With how many predictors? C) What alpha value and power values did you choose? D) What effect size did you expect? E) What was the final suggested sample size (if the power analysis was performed a-priori)? If needed, the authors can find a tutorial on power analysis here: http://doi.org/10.5334/irsp.181
- Eligibility criteria are not clear to me. Did you only select officers who were/had been COVID-19 positive?
- Paragraph 2.3.1: “in each, a high level”. What do the authors mean?
- KAP measures should be clarified: it is impossible to clearly understand what was measured. What are “practices related to COVID-19”? Are they mitigation behaviours (e.g., washing hands)? What about attitudes? This affects the ability to understand the conclusions of the paper and is reflected in ambiguity in the discussion (at p. 6 “job knowledge” is mentioned, but I wonder if that’s job-knowledge or COVID-knowledge). Maybe providing sample items for each construct, along with a brief description of the construct, could help.
- In general, I think it is redundant to repeat the same information in both text and tables. Specific values for descriptive statistics could be omitted in the text and only presented in the tables. The same applies to correlation coefficients and associated p-values. The text may simply state something like “descriptives/correlations are reported in Table x”). I would leave t-test and ANOVA results in the main text as they are now, given the absence of tables for them, and also comment the main correlations in the text (without reporting specific r values again). In summary, text and tables should be complimentary and not redundant.
- Paragraph 3.4: “a multiple regression analysis was performed using as explanatory variables age, marriage, and tenure”. The multiple regression model also included KAP constructs as independent variables (as shown in Table 4), but the text does not mention them at this point. Why is that so?
Discussion:
It is generally suitable and rooted in the findings of the paper. Only an additional suggestion:
- Being a cross-sectional study, reverse causality is always a possibility. For instance, the link between practice and job stress may work in the opposite direction (i.e., less stressed officials are able to enact more suitable behaviours/practices). I think this should be discussed and mentioned as a potential alternative explanation.
Additional comments:
- In general, minor English editing would be required, along with a careful reading-though in search of missing or duplicated words and incomplete/unclear sentences.
Author Response
Response to Reviewer 2 Comments
Abstract:
Point 1: I advise to include the sample size in the abstract.
Response 1: The subjects of this study were 375 randomly selected male correctional officials working at five South Korean correctional facilities in which COVID-19 had occurred.
Point 2: Please correct the expression “data collected data”.
Response 2: This study considered data collected with approval from 17 May 2021 to 14 June 2021.
Point 3: For instance, when mentioning preventive behavioural measures for COVID-19 mitigation (end of p. 1), you may consider citing Costantini et al. (2021; doi: 10.3389/fpsyg.2021.635406): the authors also provide some information regarding the impact of knowledge on mitigation behaviours, which may be of interest for your discussion. There are also studies investigating the impact of COVID-19 on psychological distress. I will mention some, including Viviani et al. (2021; https://doi.org/10.1016/j.future.2021.06.044), Preti et al. (2020a; https://doi.org/10.1007/s11920-020-01166-z) and Preti et al. (2020b; https://doi.org/10.1007/s11920-020-01204-w): these studies/reviews demonstrate that COVID-19 has an impact on the mental health of laypersons and specific populations, therefore they may serve as an introduction to the rationale of your own investigation.
Response 3: Based on the reference you recommended, I added the following contents.
“People who directly respond to infectious diseases can develop mental health conditions such as post-traumatic stress symptoms, depression, insomnia and severe anxiety due to high levels of job stress. These potential psychological negative consequences reduce an individual's ability to effectively handle health emergencies as well as well-being conditions [5].”
Methods:
Point 4: There are a few improvements I think should be made to the methods section.
The authors mention G-power, but no description of their power analysis is provided. For instance: A) Was it a priori? B) What kind of model did you perform the power on? A regression model? With how many predictors? C) What alpha value and power values did you choose? D) What effect size did you expect? E) What was the final suggested sample size (if the power analysis was performed a-priori)? If needed, the authors can find a tutorial on power analysis here: http://doi.org/10.5334/irsp.181
Response 4: The number of subjects was calculated by including 14 independent variables based α=.05, effect size d=.10, power value=.95 according to G*Power 3.1 program, and the minimum number of samples required for multiple linear regression was 236. Considering the dropout rate, 400 copies of the questionnaire were distributed in this study.
Point 5: Eligibility criteria are not clear to me. Did you only select officers who were/had been COVID-19 positive?
Response 5: Those who are currently working at an institution that conducted or currently conducting intensive quarantine for 4 weeks or more. due to the occurrence of COVID-19 positive, within the last 6 months.
Point 6: Paragraph 2.3.1: “in each, a high level”. What do the authors mean?
KAP measures should be clarified: it is impossible to clearly understand what was measured. What are “practices related to COVID-19”? Are they mitigation behaviours (e.g., washing hands)? What about attitudes? This affects the ability to understand the conclusions of the paper and is reflected in ambiguity in the discussion (at p. 6 “job knowledge” is mentioned, but I wonder if that’s job-knowledge or COVID-knowledge). Maybe providing sample items for each construct, along with a brief description of the construct, could help.
Response 6: This is my mistake. I modified the sentence. “All assessments used Likert scales; in each, a high level.”
In addition, an example question was added to each scale to make it easier to understand.
“For the knowledge domain(e.g.,What are the transmission routes of COVID-19?), the scale had four points (4 = ‘strongly agree’ to 1 = ‘strongly disagree’); for the attitude domain(e.g.,To what extent do you observe the preventive measures against COVID-19?), the scale had five points (5 = ‘very high’ to 1 = ‘very low’); and for the practice domain(e.g.,How often do you use a face mask?), the scale had five points (5 = ‘always’ to 1 = ‘never’).”
Point 7: In general, I think it is redundant to repeat the same information in both text and tables. Specific values for descriptive statistics could be omitted in the text and only presented in the tables. The same applies to correlation coefficients and associated p-values. The text may simply state something like “descriptives/correlations are reported in Table x”). I would leave t-test and ANOVA results in the main text as they are now, given the absence of tables for them, and also comment the main correlations in the text (without reporting specific r values again). In summary, text and tables should be complimentary and not redundant.
Response 7:
3.2. Degree of Job Stress, Knowledge, Attitude, and Practice
“Regarding COVID-19, the subjects' degree of knowledge averaged of 50.80 ± 6.54 points (range 33.0 to 68.0), the degree of attitude averaged 38.04 ± 5.46 points (range 16.0 to 50.0), the degree of practice averaged 42.29 ± 5.68 points (range 26.0 to 65.0), and the degree of job stress averaged 105.50 ± 8.97 points (range 55.0 to 132.0) (Table 2).”
3.3. Correlation between Study Variables
Knowledge, attitude, and practice were negatively correlated with each other. Job stress had a statistically significant negative correlation with knowledge (r = -0.319, p < 0.01), attitude (r = -0.127, p < 0.05), and practice (r = -0.237, p < 0.01). Knowledge had a significant positive correlation with attitude (r = 0.411, p < 0.01) and practice (r = 0.386, p < 0.01). Attitude had a significant positive correlation with practice (r = 0.408, p < 0.01) (Table 3).
Point 8: Paragraph 3.4: “a multiple regression analysis was performed using as explanatory variables age, marriage, and tenure”. The multiple regression model also included KAP constructs as independent variables (as shown in Table 4), but the text does not mention them at this point. Why is that so?
Response 8: Although "age, marridge, and tenure" were statistically significant in demographic characteristics, they were not significant in influencing factors(Table 4) and were not mentioned in the text.
Discussion:
Point 9: It is generally suitable and rooted in the findings of the paper. Only an additional suggestion: Being a cross-sectional study, reverse causality is always a possibility. For instance, the link between practice and job stress may work in the opposite direction (i.e., less stressed officials are able to enact more suitable behaviours/practices). I think this should be discussed and mentioned as a potential alternative explanation.
Response 9: I mentioned as a potential alternative explanation in discussion part.
“Considering the results of the discussion so far, the lower the knowledge and practice of correctional officials on COVID-19, the higher the level of job stress. Therefore, for the crisis management of COVID-19, it is necessary to improve the job environment related to COVID-19 of correctional officers and to raise the knowledge and practice level of COVID-19.”
